# Design of an Intermediate Die for the Multi-Pass Shape Drawing Process

**DOI:** 10.3390/ma15196893

**Published:** 2022-10-04

**Authors:** Jeong-Hun Kim, Jeong-Hyun Park, Kwang-Seok Lee, Dae-Cheol Ko, Kyung-Hun Lee

**Affiliations:** 1Department of Materials Processing, Materials Digital Platform Division, Korea Institute of Materials Science, 797 Changwon-Daero, Seongsan-gu, Changwon 51508, Korea; 2Global Research & Engineering Center, AUSTEM Co., Ltd., Incheon-si 22847, Korea; 3Department of Nanomechatronics Engineering, Pusan National University, Busandaehak-ro 63beon-gil, Geumjeong-gu, Busan 46241, Korea; 4Division of Coast Guard Studies, Korea Maritime and Ocean University, Busan 49112, Korea

**Keywords:** shape drawing, cross-roller guide, intermediate die shape design, equal-radial-velocity variation method, FE analysis

## Abstract

The multi-pass shape drawing process is mainly used in metal forming processes to manufacture long components with constant arbitrary cross-sectional shapes along their lengths. The cross-roller guide is a typical component that is manufactured by a multi-pass shape drawing process. The cross-roller guide is mostly used in optical measurement equipment where high-precision movement is required. Therefore, the dimensional accuracy of the cross-roller guide is very important since it can influence precision linear motion. However, the unfilled defects can occur in a case where the product has a complex cross-sectional shape. In this study, a new design method for an intermediate die is suggested by using an equal-radial-velocity variation method in order to reduce the unfilled defects. The proposed design method can reduce the unfilled defects by minimizing the radial velocity variation in the deformation zone of the drawing die. The intermediate die was designed by geometrical information of the final product without prior finite element (FE) analysis. The suggested method was applied to design the multi-pass shape drawing process for manufacturing the cross-roller guide. FE analysis was performed to validate the effectiveness of the proposed method in comparison to the conventional design method that uses equipotential lines in the multi-pass shape drawing process. Finally, a shape drawing experiment was performed to compare the target shape and the FE analysis with the experimental data.

## 1. Introduction

Multi-pass shape drawing is a cold forming process that is mainly used in metal forming processes to manufacture long components with constant arbitrary cross-sectional shapes along their lengths. This process provides a sleek surface finish, a precise dimensional accuracy of the cross-sectional shape, low material losses, and a low cost for tooling and mass production [1]. Lately, machine parts that have a complicated shape with grooves have been produced by shape drawing processes and are extensively used in linear motion guides, automotive components, machine tools, transport applications, industrial robots, and semiconductor manufacturing equipment. A cross-roller guide is a typical component that is manufactured by a multi-pass shape drawing process as shown in Figure 1. This component is mostly used in optical measurement equipment where high precision movement is essential. The closely controlled dimensions of the cross-roller guide can affect the precision linear motion. Therefore, designing an intermediate die is very important for a sound product in the shape drawing process [2]. An appropriate cross-sectional shape of the intermediate die minimizes the drawing load and it allows for the improvement of dimensional accuracy and uniform mechanical properties for the final product [3]. Recently, studies using the theoretical approach, experiments, and finite element (FE) simulation of the multi-pass shape drawing process have been actively conducted. Lin et al. conducted a study on the numerical and experimental investigation of trapezoidal wire drawing through a series of shape dies [4]. Oduguwa et al. performed research on the optimization of sequential process using genetic algorithms. Celano et al. performed research on the optimal design of multi-pass cold drawing processes using AI techniques [5,6]. Recently, Kim et al. proposed a virtual die design method for designing a cross-sectional shape of an intermediate die and verified the design method by performing experiments [7]. Lee applied the virtual die design method in the shape drawing process to manufacture the cross-roller guide. This method was compared with the industrial design method by evaluating the dimensional accuracy [8]. Lee designed the intermediate die shape using the half die angle to improve the dimensional accuracy in the complex shape drawing of the cross-roller guide, and he analyzed the unfilling rate and straightness that was determined by the changes of the half die angles [9]. Lee et al. proposed design method of profile using electric field theory in hot forging process [10]. Lee et al. applied the equipotential lines in the cross-sectional shape of the intermediate die in the shape drawing process to manufacture the cross-roller guide [11]. Joun et al. proposed design method for optimization of pass schedule in multi-pass extrusion and drawing by FE analysis. Sahoo et al. proposed the SERR technique for the round-to-hexagon drawing process [12].

The unfilled defects occurred along the corner of the complex cross-sectional shape in the shape drawing process. Therefore, in order to manufacture a sound cross-roller guide with a complex cross-sectional shape, the precise metal forming process is indispensable.

The objective of this study is to suggest a cross-sectional shape design method for the intermediate die that achieves high dimensional accuracy for the final product with a complex cross-sectional shape in the shape drawing process. For this study, a cause of the unfilled defects was analyzed and a cross-sectional shape design of the intermediate die in the shape drawing process was suggested by using the equal-radial-velocity variation method (ERV method) for corner filling without prior FE analysis. Afterwards, the suggested cross-sectional shape design method of the intermediate die was applied to the multi-pass shape drawing process for manufacturing the cross-roller guide, which is typical for shape drawing products. Then, the FE analysis was conducted to validate the effectiveness of the proposed method in comparison to the conventional design method that uses equipotential lines [10]. Through the FE analysis, dimensions of the cross-sectional shape of the drawn product were measured and compared between each method. Finally, an experiment for shape drawing was carried out to validate the results of the FE analysis.

## 2. Design Method for the Intermediate Die in the Shape Drawing Process

### 2.1. Equal-Radial-Velocity Variation Method (ERV Method)

#### 2.1.1. Outline of the ERV Method

In the shape drawing process, the force in the deformation zone was divided into the tensile and compressive force. The direction of the tensile force is equal to the drawing direction. The direction of the compressive force is the radial direction and the center of the material is pulled by the draw chuck. Therefore, the direction of metal flow for an arbitrary cross-sectional shape for the deformation zone is the radial direction.

In this study, the arbitrary cross-sectional shape is defined by the radius from the center to the shape at each divided section. Unfilled defects occur when the cross-sectional area of the drawn material is reduced more than the cross-sectional area of the die in the shape-drawing process. The direction of the cross-sectional area reduction is the radial direction. Consequently, the unfilled defects are related to the radial velocity distribution at each divided section in the deformation zone.

The cross-sectional area reduction and the metal flow are non-uniform in the shape drawing process to manufacture the product with a complex cross-sectional shape from the round bar because the product has a non-uniform radius at each divided section. Figure 2 shows a schematic illustration of the rod drawing process and the shape drawing process from the initial material with a round shape to the final product with a square shape. Figure 2b shows the minimum deformation distance in the radial direction from *P_5_* to *P*_6_ (*P*_5_–*P*_6_) and the maximum deformation distance in the radial direction from *P*_1_ to *P*_3_ (*P*_1_–*P*_3_). Figure 2b presents the deformation zone, which includes sections between the initial materials with the round shape and the product with a square shape.

In order to compare the sections of A and B, the schematic illustration of the deformation zone at each section is displayed in Figure 3a in two dimensions. Then, the expected path of the metal flow in the radial direction is demonstrated in Figure 3b.

The half die angle of section A is lower than the half die angle of section B. Therefore, the radial velocity of section B, *V_r,B_*, is lower than the radial velocity of section A, *V_r,A_*, for the same Z-axis velocity of sections A and B. In addition, *V_z,A_* and *V_z,B_* are identical for the equal micro deformation distance. Therefore, this drawing process provides a non-uniform metal flow at each divided section. However, the material does not separate from each other. The radial velocity of section B, *V_r,B_*, can be increased to *V_r,Bʹ_* by the radial velocity of section A, *V_r,A_*, and the metal flow can be the cause of the unfilled defects as shown in Figure 3a. Therefore, the unfilled defect occurs at the corner of the target shape by the deviation of the radial velocity in the shape drawing process as shown in Figure 3b.

#### 2.1.2. Design Procedure of the ERV Method

The ERV method can be used after deciding the number of paths. When the pass schedule is determined by the conventional method, it is possible to use this method to design the cross-sectional shape for each pass.

A shape can be expressed by the radius of the divided section with the ERV method. The ERV method is composed of an overall offset method and a local offset method. The overall offset method is used to make the preform by shape expansion in the radial direction for the same distance along the profile of the product. The local offset method was used to extend a part in the shape to reduce the radial velocity deviation due to the different half die angle.

In this study, the cross-sectional shape of the intermediate die was designed by the ERV method using only the geometric information of the products without the prior FE analysis. This was done to minimize the deviation of the radial velocity of the metal flow in the deformation zone. The design sequence of the ERV method is described as follows:

Step (1) First, the radius is drawn based on the inflection points and is drawn again since each section divided by the radius has the same angle. The radius number is represented as j=1,2…l where, l is the total number of radius.

Step (2) The geometrical information of the product is expressed as a radius, $R_{n,j}$ where, n is the number of passes.

Step (3) The radius of the minimum circumcircle is represented as a maximum radius, ${\left(R_{n,j}\right)}_{max}$ for n passes.

Step (4) The radius of the initial shape $R_{0}$ is determined by considering n and the filling rate of the material for each pass. In addition, $R_{0}$ can be calculated using Equation (1):(1)$$R_{0}={\left(R_{n,j}\right)}_{max}+k\cdot n$$
where k is the radial extension length per pass.

Step (5) The radius of the preform, $R_{i,j}$ is determined by the overall offset method that is an extension of $R_{n,j}$ in the radial direction as the same length k at each radius, which can be calculated as Equation (2):(2)$$R_{i,j}=R_{i+1,j}+k$$
where i is the pass number.

Step (6) Here, the radius is assumed to be a small, divided section and it applies the local offset method. This is determined by the area ratio (AR) of the preform area, $A_{i,j}$ to the area of the prior pass, $A_{i-1,j}$ and the average of the AR. The radius number that satisfies Equation (3) is represented as $j_{local}$, which has a relatively small half die angle:(3)$$\frac{A_{i,j}}{A_{i-1,j}}>{\left(\frac{A_{i,j}}{A_{i-1,j}}\right)}_{ave.}$$
where ${\left(\frac{A_{i,j}}{A_{i-1,j}}\right)}_{ave.}$ is the average value.

The area of each section is represented by Equation (4):(4)$$A_{i,j}=\pi R_{i,j}^{2}\theta$$

Step (7) The radius of $j_{local}$, ${R^{\prime }}_{i,j}$ is determined by ${A^{\prime }}_{i,j}$, which can be calculated by using Equation (5) with the AR:(5)$$\frac{A_{i,j}}{A_{i-1,j}}=\frac{{A^{\prime }}_{i,j}}{{A^{\prime }}_{i-1,j}}$$
where ${A^{\prime }}_{i,j}$ is the area of the small, divided section after the local offset and ${A^{\prime }}_{i-1,j}$ is a fictitious value where the overall offset method is applied from $A_{i-1,j}$ and ${A^{\prime }}_{i,j}$. This is determined differently by the AR according to each small, divided section. Thus, the profile of the intermediate die is derived from the $R_{i,j}$ and ${R^{\prime }}_{i,j}$. Finally, the profile of die entry and exit of each pass is determined by the ERV method, and the half die angle is automatically determined.

### 2.2. Die Design of the Cross-Roller Guide

The ERV method was applied to the multi-pass shape drawing process for the cross-roller guide. The geometric information of the cross-roller guide in Figure 1 is necessary to design the cross-sectional shape of the intermediate die. In Step (1), the cross-roller guide has one plane of symmetry. The cross section of the product was divided with a small angle (***θ***) of 8° as much as the fan-shape, basically. Then, the angle which expresses the corner was modified. The modified angle was considered to determine the area of each section. Thus, the radius number, l, was 23 to simplify the calculations even though it is possible to increase the number of radii for a smooth shape. Thus, the radius number is represented as j=1,2…23, as shown in Figure 4, and the number of passes, n, is two in this study. In Step (2), the geometrical information of the cross-roller guide is expressed as a radius, $R_{2,j}$ as show in Figure 5b. In Step (3), the radius of the minimum circumcircle, ${\left(R_{2,j}\right)}_{max}$ is 7 mm for the radius numbers 7 and 17 are demonstrated in Figure 4. In Step (4), the radial extension length per pass, k is 1 mm [10]. The radius of the initial shape $R_{0}$ was assumed as 9 mm, which can be calculated from Equation (1) as shown in Figure 5a. In Step (5), the radius of the preform is $R_{1,j}$, which is calculated as $R_{2,j}+1$ based on Equation (2). In addition, Figure 5b shows the preform of the intermediate die for the cross-roller guide between the initial shape and the target shape. In Step (6), the value of the intermediate pass number, i=1, is substituted from Equation (1) into Equation (2).
(6)$$\frac{A_{1,j}}{A_{0,j}}>{\left(\frac{A_{1,j}}{A_{0,j}}\right)}_{ave.}$$

Therefore, the radius number, $j_{local}$ ranges from 4 to 9 and 15 to 20, which satisfies Equation (6). In Step (7), The radius of $j_{local}$, ${R^{\prime }}_{1,j}$ is determined by ${A^{\prime }}_{1,j}$ which can be calculated as Equation (7) using the AR.
(7)$$\frac{A_{1,j}}{A_{0,j}}=\frac{{A^{\prime }}_{1,j}}{{A^{\prime }}_{0,j}}$$

Table 1 shows the value of $R_{2,j}$, $R_{1,j}$, $A_{1,j}/A_{0,j}$, ${\left(A_{1,j}/A_{0,j}\right)}_{ave.}$, and ${R^{\prime }}_{1,j}$ of the cross-roller guide. Figure 6 shows the intermediate die shape of the cross-roller guide, which is determined by using the ERV method and the shape of the second pass die is equal to the target shape. ${R^{\prime }}_{1,j}$ has a maximum value when j is 7 or 17. Thus, when j is 7 or 17 and k is 0.9, $R_{0}$ is 8.8 mm and ${R^{\prime }}_{1,7}$ is 8.9 mm. When j is 7 or 17 and k is 0.8, $R_{0}$ is 8.6 mm and ${R^{\prime }}_{1,7}$ is 8.91 mm. Therefore, when k is lower than 1, an undercut occurs and the size of the initial round bar, ø17.6 and ø17.2, were not manufactured with a commercial material. It is easy to get the size, ø18, of the initial rod for the shape drawing experiment when k is 1. Therefore, the initial rod was determined as 9.1 mm, which is larger than maximum value of ${R^{\prime }}_{1,j}$.

## 3. Verification of the Proposed Design Method

### 3.1. FE Analysis Model

In order to verify the effectiveness of the intermediate die shape of the cross-roller guide that was designed using the ERV method, FE analysis was performed with the intermediate die shape designed by using the equipotential line method. This is a conventional method in the shape drawing process. Figure 7 shows the die shape which was designed as the ERV method for the FE analysis. Figure 8 illustrates the die shape, which was designed as an equipotential line method. A comparison of the ERV method and the equipotential line method for the cross-sectional shape in the two-pass shape drawing is demonstrated in Figure 9.

The FE analysis was performed using the DEFORM-3D Ver.10.0 software (Columbus, OH, USA) environment. The initial material was AISI 1020 with an 18.2 mm diameter that was applied to the FE analysis in the two-pass shape drawing process, and initial temperature was considered as room temperature. The flow stress equation of the annealed material was used in the shape drawing experiment as described in Equation (8):(8)$$\overset{¯}{\sigma }=703.07\cdot {\overset{¯}{\varepsilon }}^{0.227}\left(\mathrm{MPa}\right)$$

The FE simulation of the shape drawing process was conducted with a rigid-plastic FE model to reduce the computation time, which was enabled by the plane symmetrical configuration of the workpiece as shown in Figure 10. The initial mesh structure of the FE model was constructed with approximately 100,000 initial tetrahedral elements, and an automatic remeshing scheme was used in the numerical simulations. The die tip was considered as rigid, and the workpiece was pulled along the drawing direction. The length of the initial material was set to 50 mm in consideration of the computation time. The friction factor (m) was fixed as 0.1 between the material and die; the drawing speed (*v_d_*) was 25 mm/s. The pre-strain of the drawn material from the first pass was set to zero. This is due to the annealing prior to entering the second pass in the shape-drawing experiment. The dimensional accuracy of the drawn product from the FE analysis was evaluated by the unfilling rate (*UR*), which was calculated using Equation (9):(9)$$UR=\left(1-\frac{A_{analysis}}{A_{target}}\right)\times 100\;\left(\%\right)$$
where $A_{target}$ is the area of the target shape, and $A_{analysis}$ is the area of the FE-simulation result. A lower *UR* indicates that the dimensional accuracy is excellent.

The total reduction area (RA) is 59.3% of the cross-roller guide. The RA of the first pass and the second pass is 34% and 38.5%, respectively, in the case of the ERV method. The RA of the first pass and the second pass is 36.3% and 36.3%, respectively, in the case of the equipotential line method.

### 3.2. Results of the FE Analysis

Figure 11 illustrates the contours of the effective strain in the two-pass shape-drawing process at room temperature. Material deformation becomes uniform in the case of the ERV method because the radial velocity variation is minimized in the deformation zone of the second pass drawing die.

Figure 12 depicts the comparison of the cross-sectional shape of the drawn material and the target shape from the second pass FE analysis results for each design method of the cross-roller guide. The UR in the case of the ERV method is lower than the case of the equipotential line method. This is because the distribution of the effective strain and the radial velocity are more uniform in the case of the ERV method than the case of the conventional design method.

Figure 13 shows the comparison of the radial velocity at the deformation zone in the case of the ERV method and the equipotential line method. The horizontal axis is the deformation distance of the drawing direction and the vertical axis is the radial velocity. The radial velocity deviation in the case of the equipotential line method is higher than the case of the ERV method. This is because the difference of the half die angle is higher than the case of the ERV method. This can be the explanation of the FE analysis results for the UR. Table 2 shows the comparison of the RA, drawing load, and the UR from the FE analysis for each design method of the cross-roller guide.

Therefore, it was verified through the FE analysis for the cross-roller guide that the radial velocity is different according to the half die angle in the deformation zone. This can be the cause of the unfilled defect and it is possible to reduce the UR by the ERV method.

## 4. Experiment of the Two-Pass Shape-Drawing Process

The multi-pass shape drawing experiment was conducted to validate the effectiveness of the ERV method that was proposed in this study. Figure 14 shows the drawing dies that were designed for the multi-pass shape drawing experiment to manufacture the cross-roller guide.

### 4.1. Condition of the Experiment

The initial material was AISI 1020 with an 18 mm diameter and a 600 mm length. After the pointing process was conducted, the annealed material was fixed to the drawbench through the first pass die. The annealed material was lubricated and the first pass shape-drawing process was performed at room temperature. The annealing process was carried out after the first pass shape-drawing process to remove the residual stress. The second pass shape-drawing process was performed after the annealed material was treated by the pointing process and lubrication. The drawing speed in the experiment was almost 25 mm/s, which is the same that was used for the FE analysis in the shape-drawing process.

### 4.2. Experimental Results and Discussion

The materials that were drawn for each pass and the cross-sectional shapes of the materials are depicted in Figure 15. The dimensions of the height and width were measured with digital Vernier calipers. These measurements were within the dimensional error that is presented in Figure 1 and the maximum dimensional error was 0.02 mm. The cross-sectional area of the specimen was collected from the drawn material. This was measured by using a 2D vision measuring machine (SMART2010, VMS 3.1 measuring S/W, China). The maximum dimensional error between the profile of the specimen and the target shape was measured by using a vertical section profile projector (NIKON V-12BDC, Japan). This was done to evaluate the dimensional accuracy after the multi-pass shape drawing experiment. The UR was calculated by using the cross-sectional area of the specimen and the target shape, which was measured by using the 2D vision measuring machine and the value of the UR was less than 1%. Figure 16 displays a cross-sectional shape comparison of the specimen, target shape, and the FE analysis results. It also shows the dimensional error (➀, ➁, ➂, ➃) between the profile of the specimen and the target shape. Each dimensional error was −0.077, −0.064, −0.079, and −0.069, respectively.

Table 3 lists the comparison of the UR and the maximum value of the dimensional error for the specimen and the target shape between the experiment and the FE analysis in the cross-roller guide.

The FE analysis and the experimental results at each pass displayed the same trend. The experimental results satisfied the allowable tolerance, which was ±0.1 mm for the cross-roller guide. Therefore, the proposed ERV method that was used to design the cross-sectional shape of the intermediate die was applied to the multi-pass shape drawing process. This consists of a two-pass for manufacturing the cross-roller guide and the method was verified by manufacturing the product with a sound shape, which was within the required tolerance.

## 5. Conclusions

In this study, a cause of unfilled defects was analyzed and the cross-sectional shape design method of the intermediate die, equal-radial-velocity variation method (ERV method) is suggested. In order to verify the effectiveness of the suggested method, FE analysis was performed with the intermediate die shape designed by using the equipotential line, which is a conventional method. Then, the multi-pass shape-drawing experiment was conducted to validate the effectiveness of the ERV method and results were compared with the FE analysis. The main conclusions are as follows:The cause of the unfilled defects that occurred by the deviation of the radial velocity was the different half die angles along the profile between the initial material and the final product. The design method of the cross-sectional shape of the intermediate die by using the ERV method can reduce the unfilled defects. This can be achieved by minimizing the radial velocity variation in the deformation zone of the drawing die. The results from this research provide a valuable guideline for the intermediate die design in the shaped drawn production.The FE analysis was conducted to validate the effectiveness of the proposed method. It was also compared to the radial velocity in the deformation zone with the conventional design method using equipotential lines for manufacturing the cross-roller guide. Compared to the conventional design method, it has been verified that the radial velocity variation in the deformation zone of the proposed ERV method is minimized. By minimizing the radial velocity variation, the UR of the cross section could be reduced from 1.064% to 0.56%.The shape drawing experiments were carried out to verify the validity of the suggested method by using dies for manufacturing the cross-roller guide. The ERV method led to a successful bar shape and the highest dimensional precision. The UR was 0.69%, and the dimensional errors between the profile of the specimen and the target shape were measured as 0.064 mm and 0.079 as minimum and maximum, respectively.

## Figures and Tables

**Figure 1 materials-15-06893-f001:**
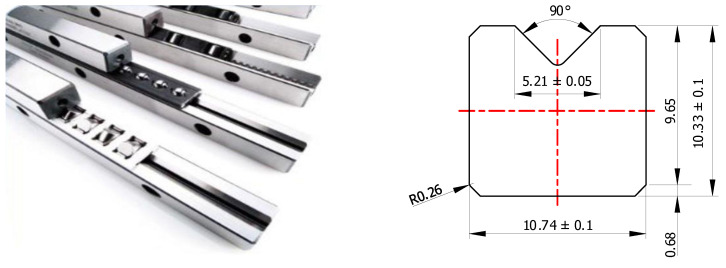
Photograph and dimensions of the cross-roller guide (unit: mm).

**Figure 2 materials-15-06893-f002:**
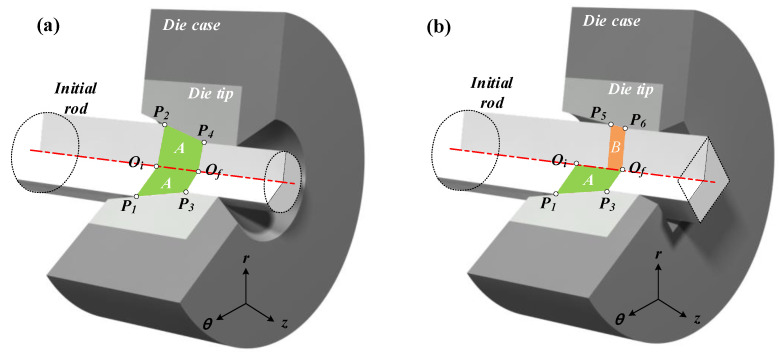
Schematic of the deformation zone in the drawing process: (**a**) rod-drawing process and (**b**) shape-drawing process.

**Figure 3 materials-15-06893-f003:**
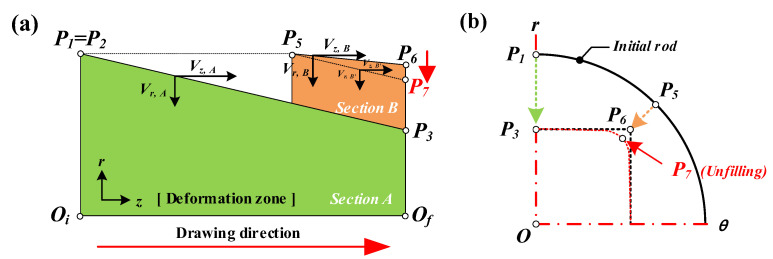
Expected path of metal flow: (**a**) comparison of sections A and B for the shape drawing process and (**b**) metal flow in radial direction.

**Figure 4 materials-15-06893-f004:**
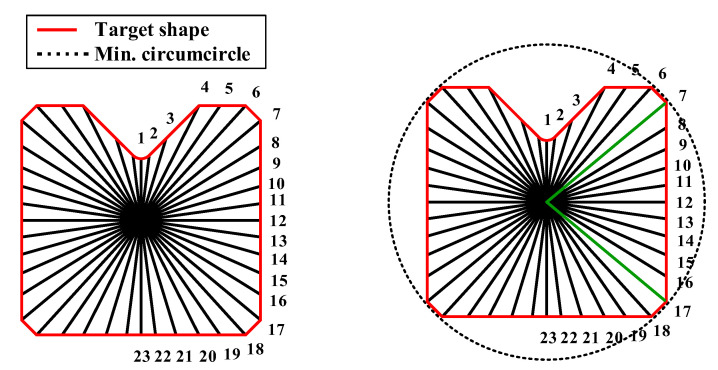
Radius from the center to the target shape of the cross-roller guide.

**Figure 5 materials-15-06893-f005:**
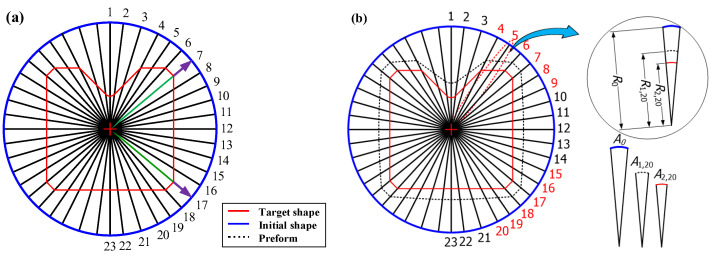
(**a**) Initial and target shapes of the cross-roller guide and (**b**) preform of the cross-roller guide.

**Figure 6 materials-15-06893-f006:**
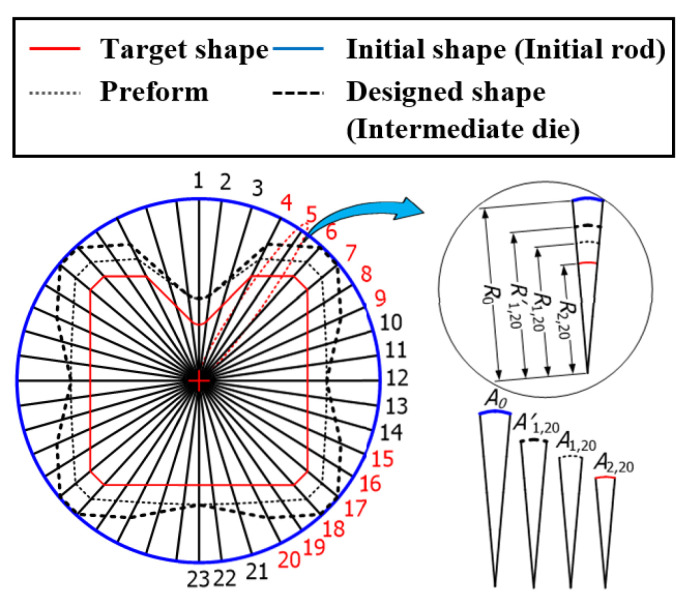
Intermediate die shape of the cross-roller guide.

**Figure 7 materials-15-06893-f007:**
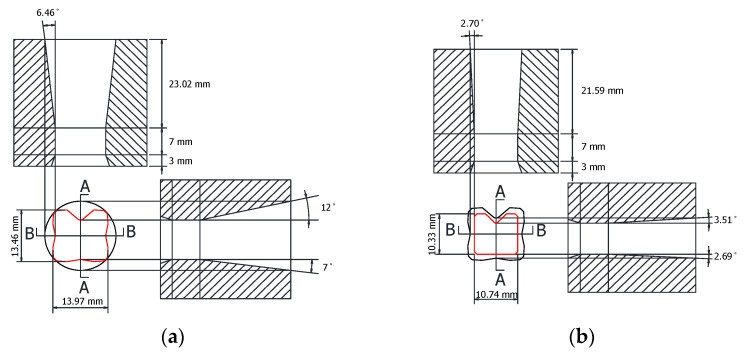
Geometry of the drawing dies designed using the ERV method for the cross-roller guide: (**a**) first pass; (**b**) second pass.

**Figure 8 materials-15-06893-f008:**
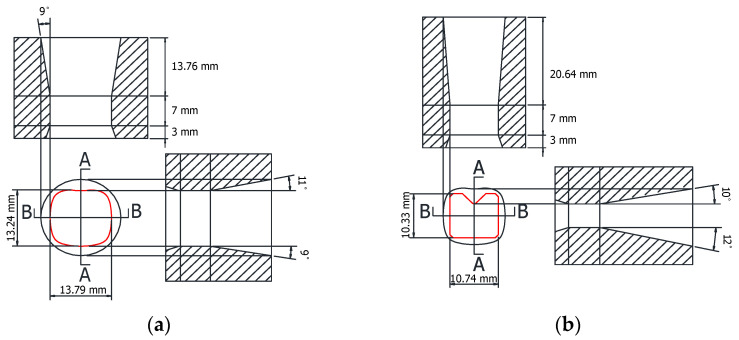
Geometry of the drawing dies designed using the equipotential line method for the cross-roller guide: (**a**) first pass; (**b**) second pass.

**Figure 9 materials-15-06893-f009:**
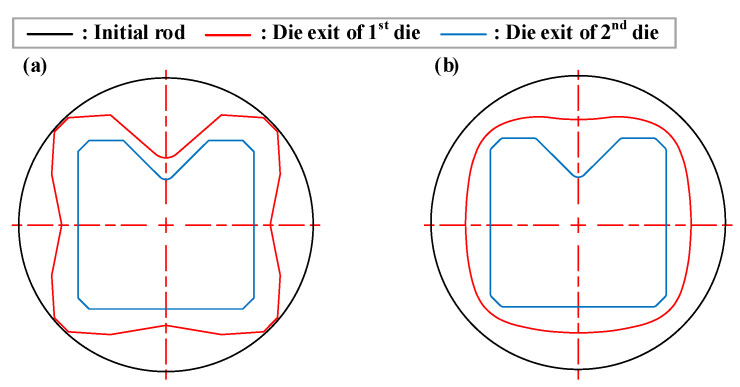
Cross-sectional shapes of the two-pass shape drawing dies: (**a**) ERV method; (**b**) equipotential line method.

**Figure 10 materials-15-06893-f010:**
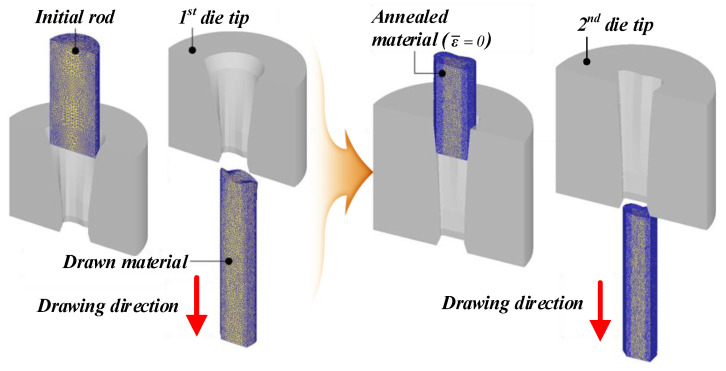
FE model of the shape-drawing process for the cross-roller guide.

**Figure 11 materials-15-06893-f011:**
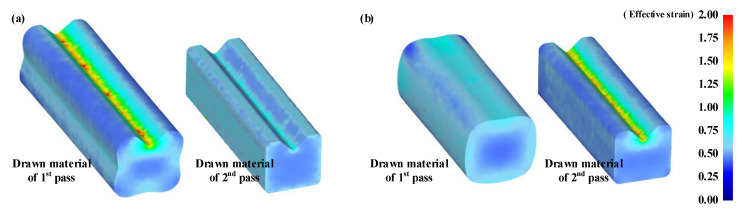
FE model of shape drawing process for the cross-roller guide: (**a**) ERV method; (**b**) equipotential line method.

**Figure 12 materials-15-06893-f012:**
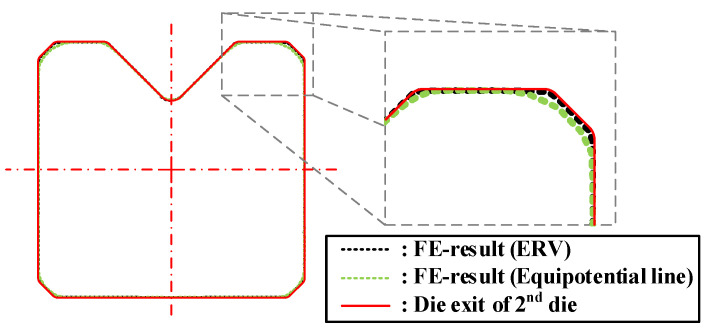
Comparison of the cross-sectional shapes of the second pass for the cross-roller guide.

**Figure 13 materials-15-06893-f013:**
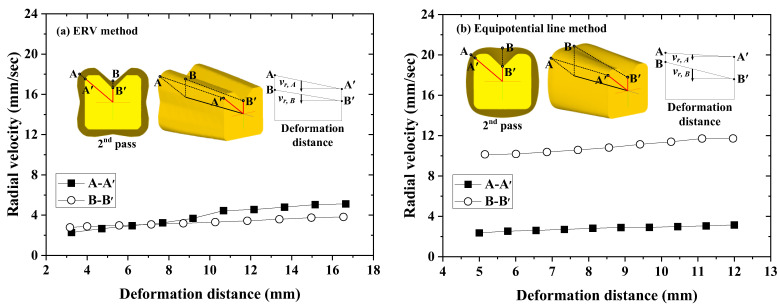
Comparison of the radial velocity of the second pass for the cross-roller guide: (**a**) ERV method; (**b**) equipotential line method.

**Figure 14 materials-15-06893-f014:**
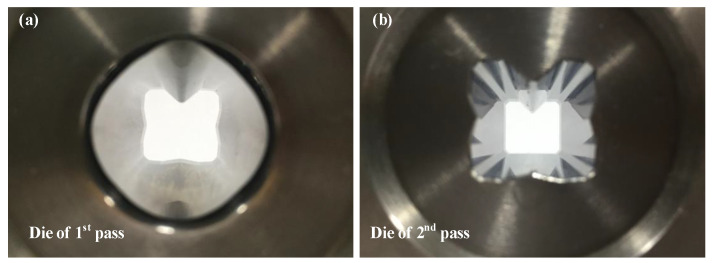
Drawing dies of the multi-pass shape drawing experiment for the cross-roller guide: (**a**) die of first pass; (**b**) die of second pass.

**Figure 15 materials-15-06893-f015:**
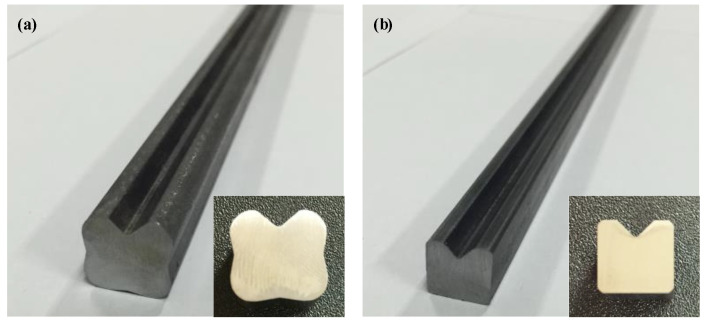
Drawn material in each pass for the cross-roller guide: (**a**) first pass; (**b**) second pass.

**Figure 16 materials-15-06893-f016:**
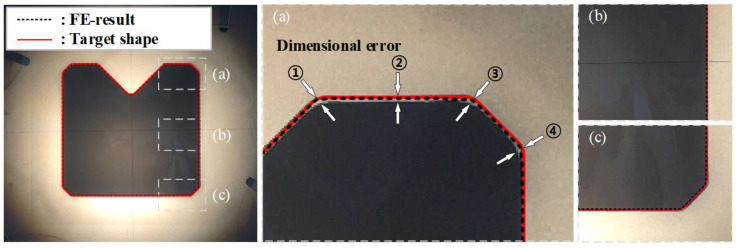
Comparison of the cross-sectional shape for the cross-roller guide: (**a**) Top; (**b**) Middle; (**c**) Bottom.

**Table 1 materials-15-06893-t001:** ERV method for the cross-roller guide (${\left(A_{1,j}/A_{0,j}\right)}_{ave.}=0.56).$

*j*	$R_{2,j}\;(\mathrm{mm})$	$R_{1,j}\;(\mathrm{mm})$	$A_{1,j}/A_{0,j}$	${R^{\prime }}_{1,j}\;(\mathrm{mm})$
1	2.78	3.78	0.18	4.28
2	2.95	3.95	0.19	4.48
3	3.85	4.85	0.29	5.51
4	5.78	6.78	0.57	7.71
5	6.27	7.27	0.65	8.27
6	6.98	7.98	0.79	9.08
7	7	8	0.79	9.10
8	6.33	7.33	0.66	8.33
9	5.88	6.88	0.58	7.82
10	5.59	6.59	0.54	7.49
11	5.43	6.43	0.51	7.30
12	5.37	6.37	0.5	7.24
13	5.43	6.43	0.51	7.30
14	5.59	6.59	0.54	7.49
15	5.88	6.88	0.58	7.82
16	6.33	7.33	0.66	8.33
17	7	8	0.79	9.10
18	6.98	7.98	0.79	9.08
19	6.27	7.27	0.65	8.27
20	5.78	6.78	0.57	7.71
21	5.4	6.4	0.51	7.27
22	5.2	6.2	0.48	7.05
23	5.15	6.15	0.47	7.01

**Table 2 materials-15-06893-t002:** Results of the FE analysis for the cross-roller guide.

		ERV Method		Equipotential Line Method	
		First Pass	Second Pass	First Pass	Second Pass
RA (%)		35.91	36.98	37.36	35.87
Drawing load (tons)		5.00	2.74	4.54	2.84
UR (%)		2.96	0.56	1.67	1.064

**Table 3 materials-15-06893-t003:** Comparison of the UR and dimensional error for the cross-roller guide $(A_{target}=103.33\;{\mathrm{mm}}^{2}).$

		FE Analysis		Experiment	
		First Pass	Second Pass	First Pass	Second Pass
Area (mm^2^)		163.05	102.75	161.09	102.6
UR (%)		2.96	0.56	4.23	0.69
Max. dimensional error (mm)			−0.036		−0.079

## Data Availability

Data sharing is not applicable to this article.

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
