# Peer review of "Design of an Intermediate Die for the Multi-Pass Shape Drawing Process"

_materials, 2022, doi:10.3390/ma15196893_

Round 1
Reviewer 1 Report
The presented work proposes a solution of great interest to solve the problems of lack of filling of the material when complex profiles are processed by cold drawing process. The proposed design method has proven to be effective for this purpose and corroborating numerical and experimental results have been presented. However, a series of improvements must be made before the work can be published. To do this, I ask you to consider the following questions in a new version of the manuscript:
1.- Consider to add the information about the temperature conditions of the process, both for the FE model and in the experiment.
2.- Add information and justification about the configuration of the parameters and options for the mesh generation of the initial rod.
3.- Add information about the type of objects that have been assumed for the FE model.
4.- General conclusions have been presented on the design proposal. However, specific conclusions should be added that assess the results obtained, both from the FE simulations and from the experimental process, which are shown in tables 1 and 2.
5.- A specific assessment of the results of each case and a comparative assessment, for example, would be interesting.
These questions are noted in the PDF version of the original manuscript.
The Reviewer.

Reviewer 2 Report
General assessment
This is a commendable study of pass design in complex section drawing. The subject is undeniably important. The validity of the simple design approach proposed is proved by comparison with FEM simulations. The interest of this new design method is underlined by the comparison with an existing one (Equi-Potential Line method, EPL). The paper structure and style are good. However the precision of the information could be improved, in particular on the complex, multistep procedures used to approach the optimal die shape. Remember a scientific paper must be written in such a way that an experienced reader should be able to reproduce the work – and check it.
Detailed questions and remarks
- Since this is precision manufacturing and in order to better put forward the improvement brought by the new design strategy, I think it would be useful to indicate tolerances along with dimensions in fig.1, or at least to tell if all dimensions have the same functional importance for the application. A value of ±0.1 mm is given but just at the end of the paper, is it the same for all dimensions?
- A lot of details are given on how to design the shape of the die hole. However, in the examples, one sees the half die angle is “j-section-dependent” and strongly so – as can be seen in fig.7 for the ERV and fig.8 for the EPL. I think this is necessary to maintain all entry points in roughly the same plane (is this the aim?). In this case, this is an integral part of the design strategy and it should be mentioned.
Maybe this is alluded to in the bibliographic part, p.2 lines 62-63 “changes of the half die angles at each section”: what does “section” mean here? Each pass, or each longitudinal section j as defined later? See also p.3 line 114. Please clarify, I think this half-die-angle circumferential distribution is central for the success of the design method (since “equal radial velocity” is desired).
- Section 2.1.2 is difficult to follow.
First, please make it clear that the steps you describe are better understood using figs 4-6 (no call to these figures is made here, only in section 2.2).
Second, for the “intermediate variable” Ri,j you use the term “preform” whereas the “finally designed shape” is called “intermediate” in fig.6 : you probably mean the shape after the intermediate pass, I can understand, but this terminology is confusing.
The relationship between R’i,j and A’i,j should be given.
I see A’0,j , are they all identical = A0 ? Is the point distribution equi-angular?
- I have a problem between table 1 and fig.7, which shows I could not follow properly the design strategy. In table 1, R’1,j is exactly 11.1% more, whatever j, than R1,j = R2,j (target) + 1 mm. This is absolutely not the case in fig.7 between “intermediate” (R’1,j?) and “preform” (R1,j)? Did I miss one step? I can understand the column A1,j/A0,j = (R1,j/R0)**2, but not the passage to R’1,j.
- Section 3, line 218, a detail: the simulation is symmetric with respect to a longitudinal plane, not axisymmetric.
- For my curiosity, is the intermediate annealing an integral part of the final process, or just introduced for the experiments carried out for this paper? In the former case, eliminating plastic strain between pass 1 and pass 2 is relevant (fig.11). The superiority of the ERV lies in great part in leaving only quasi-uniform strain to perform to obtain the shape after pass 2 ; for the same reason the filling defect is much smaller.
Otherwise, strains of pass 1 and 2 should be cumulated and the maps by ERV and EPL would look much more the same, I guess.
- Fig.16: is it possible to draw this comparison at larger magnification? Right now, it is impossible to see anything, even at large magnification of the PDF file. Maybe try a representation with gap amplification, rather than an isometric cross section.
- Bibliography: 9 references out of 11 are from the same team : one author at least in common with the present paper. I am aware of other papers on section drawing pass schedule optimisation, which are not mentioned here. Please diversify your literature. It is generally considered that 1/3 of self-citations is OK – it may of course depend on the subject and the number of teams working on it, but well, 80%...
